# Data Curation for Image Captioning with Text-to-Image Generative Models

## Abstract

Recent advances in image captioning are driven by increasingly larger-scale vision–language pretraining, relying on massive computational resources and increasingly large datasets. Instead of solely focusing on scaling pretraining, we ask whether it is possible to improve performance by improving the quality of the samples in existing datasets. We pursue this question through two approaches to data curation: one that assumes that some examples should be avoided due to mismatches between the image and caption, and one that assumes that the mismatch can be addressed by replacing the image, for which we use the state-of-the-art Stable Diffusion model. These approaches are evaluated using the BLIP model on the COCO and Flickr30K datasets. Models trained with our data curation approaches consistently outperform their baselines, indicating that better image captioning models can be trained by curating existing resources. Finally, we conduct a human study to understand the errors made by the Stable Diffusion model and highlight directions for future work in text-to-image generation.

## 1 Introduction

Large-scale vision–language pretraining has been the driving force behind recent advances in image captioning [14]. The amount of image–text data needed to pretrain recent generative language models [28, 23, 53] has made it necessary to train on "noisy" samples harvested from the web [46, 45], as opposed to crowdsourced captions [32]. This emerging reliance on harvested data has made it important to perform additional filtering steps to remove low-quality data [28], in addition to more resource-intensive pretraining. Given that computing resources are not equally distributed [21], there is a need to also pursue less resource-intensive research directions.

We show how to improve image captioning by improving the quality of the downstream task data through *data curation*: the process of dynamically updating the samples during training. We devise three techniques for data curation that are designed to prevent the total size of the dataset from increasing: the complete removal of an image–caption sample from a dataset; replacing a caption with another caption; and replacing images using a text-to-image generation model [41]. These curation techniques are used to update image–caption samples that have outlier losses, with respect to the rest of a training dataset, under the current model parameters. In other words, the samples that are proving *difficult* to model. Also, the synthesis of completely new images is radically different from standard data augmentation techniques, such as random cropping or color manipulation [47], or swapping and mask words in text [12].

We conduct experiments using BLIP [28], a strong image captioning model, on the Flickr30K [56] and MS COCO datasets [32]. The results show that the sample removal and image replacement techniques lead to consistent improvements of 1–3 CIDEr points compared to not curating the dataset. Our analyses show that Flickr30K benefits from more curation than COCO due to differences

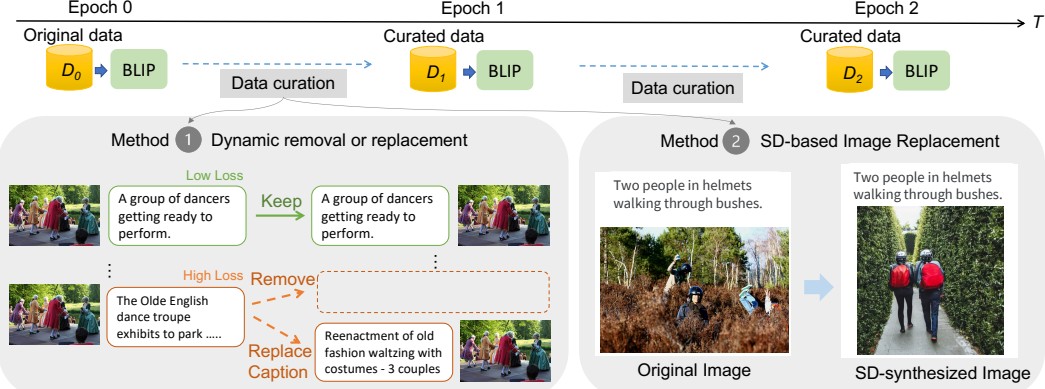

Figure 1: Overview of our data curation approaches. For dynamic removal or replacement of captions, high loss image-text pairs are either removed or the image is paired with an alternative caption in the following training epoch. For image replacement, captions of original images are used as prompts for text-to-image generation to synthesize new image–text pairs. We experiment with both options of replacing the image only, or pair another relevant caption to the synthesized image.

in the distribution of long captions in each dataset. Finally, we find that it is better to curate the data dynamically while training instead of replacing images before starting to train the model. Taken together, these findings show the promise of *model-in-the-loop* text-to-image generation for multimodal learning, while highlighting that improvements in text-to-image generation are likely to further enhance the effectiveness of data curation.

## 2   Related work

**Image Captioning**   Image Captioning is the task of describing images with syntactically and semantically sentences. Current deep learning-based image captioning models have evolved as the encode-decoder frameworks with multi-modal connection [8, 9], attentive [24, 16] and fusion strategies [58]. Standard captioning datasets contain Flickr30K [56] and the commonly used MS COCO [32], which consisting of images with events, objects and scenes. Each image is paired with five captions. Some works have demonstrated the benefits of training on synthetic captions [29, 3] or datasets collected from other vision-and-language learning tasks [38, 7].

**Data Augmentation**   Data augmentation [13] has achieved increasing attention in both natural language processing [33] and vision-and-language learning [27]. Early methods generate augmented examples in the model's feature space [54] or interpolate the inputs and labels of few examples [57]. For downstream tasks in the text domain, Yang et al. [55] and Anaby-Tavor et al. [1] generate synthetic text examples through state-of-the-art pretrained language models and show improved performance on common-sense reasoning and text-classification. For image captioning, BERT [11] has been used to generate additional captions to improve the diversity of the captioning datasets [3]. Hossain et al. [22] used GAN-synthesized images as additional augmentation training set to improve image captioning models.

**Diffusion Models and Application**   Diffusion models [49, 35] have grown rapidly and become the powerful deep generative models. They have shown potential in a variety of applications, including text-to-image generation [36, 15], image-to-image translation [42], as well as semantic segmentation [26, 5] and video generation [20, 48, 52]. While recent large scale latent diffusion models have shown strong capability in generating both artistic and photo-realistic high-resolution images [41, 34, 39, 43], applying large-scale stable diffusion models in vision-language downstream tasks remains under-explored. Concurrently, Azizi et al. [4] and Jain et al. [25] show that image classifiers can be improved by learning from augmentation images generated by finetuned stable-diffusion models. To the best of our knowledge, we are the first to explore how image captioning models can benefit from simple data curation without scaling up existing datasets, and how stable-diffusion text-to-image models can be applied and contribute in the process.

## 3   Data Curation for Captioning

Our goal is to improve image captioning models by preventing the model from training on difficult samples. There are many reasons for the possible existence of these difficult samples, including mismatches or inconsistencies between the image and caption [3]. More formally, given an image captioning training dataset $\mathcal{D}$ with $K$ images, let $I_k$ be the $k$-th image. Each image is paired with $J$ captions; let $C_k^j$ be $j$th caption of image $k$, and thus, let $(I_k, C_k^j)$ be an image–caption sample in the dataset. Assume the existence of model $\mathcal{M}$, which is being trained on dataset $\mathcal{D}$, from which we can calculate the loss of each sample at each epoch $t$: $\mathcal{L}_{\mathcal{M}}^t(I_k, C_k^j)$, which can be used to track the difficult samples. At the end of each epoch, the difficult samples are candidates for our data curation techniques, resulting in dynamic updates to the training dataset $\mathcal{D} \to \mathcal{D}_1 \to \cdots \to \mathcal{D}_T$.

### 3.1   Identifying the difficult samples

Difficult training samples may contain mismatches or inconsistencies between the image and the caption [3]. We propose to use the captioning model that is being trained to automatically identify such samples. After each epoch, we compute the loss of each sample in the current training dataset, given the current model parameters. The highest loss samples are targets for our data curation methods; more specifically, we focus on samples with losses that are either two standard deviations from the mean, or a fixed X% away e.g. 10%, 20%, etc. In this way, the training dataset is dynamically updated at the end of each epoch according to the model's captioning capability. The adjacent figure shows the empirical distribution of losses in the training samples of the Flickr30K dataset. It is clear that, without data curation, the high-loss samples remain high-loss during five epochs of training.

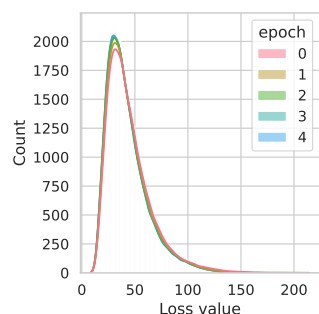

Figure 2: Distribution of per-sample losses in Flickr30K.

### 3.2   Sample Removal / Caption Replacement

The simplest approach to data curation is to remove or replace the high-loss samples. In REMOVE, the high-loss samples are completely removed from the remainder of the training process, reducing the total number of image–caption training samples. In REPLACECAP, we simply replace the caption in the image–caption sample with a different caption taken from the other captions that describe the image, effectively creating a duplicate. With the caption replacement method, the total number of samples used to train the model remains the same, as well as the total number of the unique images. This creates a clean control condition for the subsequent experiments.

### 3.3   Image Generation-based Replacement

An alternative to removing difficult samples or replacing captions is to pair an existing caption with a new image. This has the benefit of training the model on the same total number of samples while exposing it to more unique images. The new image could be found by humans, in a long-running human-in-the-loop cycle. Instead, we use a text-to-image generation model, in a rapid model-in-the-loop step, to synthesize images based on the other sentences that describe the image. Some representative examples of images generated using this technique can be seen in Figure 10.

Our methodology is based on the open source Stable Diffusion model [41], which can generate images given a textual prompt.[1] We integrate this into training as follows: Given an image $I_k$ in the training data and its captions $\{(I_k, C_k^1), \ldots, (I_k, C_k^J)\}$, we synthesize a new image $\hat{I}_k$ without increasing the total number of samples in the original dataset. Instead, we replace the original image in the sample with the generated image. Specifically, for image $I_k$, we replace a high-loss sample $(I_k, C_k^j)$ with the synthesized image-text pair $(\hat{I}_k, C_k^j)$.

---

[1]It is also possible to use API-based models but we chose Stable Diffusion for two reasons: (i) Stable Diffusion can be integrated directly into our training pipeline using the open source code. And (ii) we estimate that it would cost $7,424 to run a single experiment on the Flickr30K dataset using DALLE-2.

**Round-trip captioning evaluation**

In order to effectively use a text-to-image generation model for data curation, we need an objective measure that can estimate the expected quality of a generated image. Most previous work uses image-oriented measures like FID [19] or CLIPScore [17] but these measures are claimed to lack alignment with perceptual quality [44]. We also found they were not suitable for our purpose, and that CLIPScore cannot distinguish between low- and high-loss samples in the captioning model (Figure 9). Here, we propose an alternative that is directly related to our task: given the generated image, measure the quality of the caption that can be generated by a fixed model.

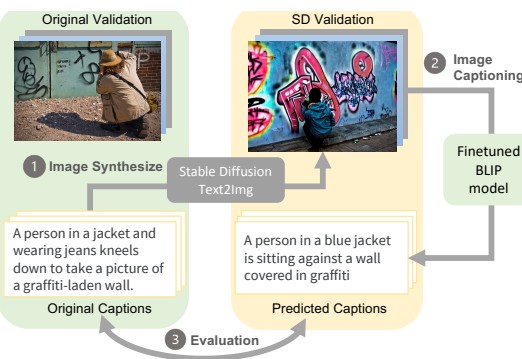

Our assumption is that if the generated images are of a similar quality to the original images, the resulting captions should be similar to each other. We call this a round-trip captioning evaluation, which comprises three steps illustrated in Figure 3. In Step (1), we use the captions in the validation set to generate images using a text-to-image generation model. In Step (2), we use an existing image-captioning model to predict captions for the generated images. Specifically, we use BLIP fine-tuned on the COCO dataset but any other strong captioning model could be used instead. Finally, in Step (3), we compare the predicted captions against the original captions. We now discuss the the factors that we found make a difference when generating images.

Figure 3: Round-trip captioning evaluation.

**Prompt engineering matters**

Recall that text-to-image generation models produce images based on a textual prompts. Given a set of five captions that describe an image, there are several options for how to prompt the image generation model. We experiment with three options:

- Single caption: Each caption is used in isolation to generate a new image.

- Sentence-BERT selection: There is a lot of variety in how different captions describe the same image. Instead of using all captions, we can use a representative caption from the set. This is achieved using the Sentence-BERT [40] model to find the caption that is closest to the average embedding of all captions.

- Concatenation: All five captions are concatenated as the text prompt for generation.

For all three approaches mentioned above, we can append an additional string to the prompt as a *styler* to force a specific style in the generated image (+Styler). The styler used here is: "national geographic, high quality photography, Canon EOS R3, Flickr".[2]

**Finetuning improves image relevance**

Table 1 shows the results of the round-trip captioning evaluation on the Flickr30K dataset using different textual prompts and whether or not to fine-tune the diffusion model. When we fine-tune StableDiffusion, we use the MS COCO [32] dataset with a prompt consisting of a concatenation of all 5 captions, for 15,000 steps with a constant learning rate of $1e-5$ and a batch size of 32. The best performance is clearly found by fine-tuning Stable Diffusion 1.5 and using a prompt with a concatenation of the captions and the styler. We use this configuration in the remainder of the paper.

Table 1: Round-trip captioning evaluation on Flickr30K with different Stable Diffusion models, prompts, and fine-tuning. **B**LEU, **C**IDEr, **M**eteor.

| Model | FT | Prompt | B | C | M |
|---|---|---|---|---|---|
| Upper-bound | | | 37.6 | 27.2 | 57.1 |
| SD 1.5 | - | concat | 31.0 | 24.7 | 52.5 |
| SD 1.5 | - | + styler | 30.8 | 24.2 | 52.5 |
| SD 1.5 | F | + styler | **33.5** | **25.0** | **53.5** |
| SD 1.5 | F | SBERT + styler | 30.6 | 24.1 | 52.0 |
| SD 2.0 | - | concat + styler | 31.2 | 24.8 | 52.0 |

---

[2]The styler was chosen by inspecting the generated images, with a preference against "artistic" outputs.

Table 2: Results for standard finetuning with data curation. We find improvements for all curation methods compared to the baseline of training on the original datasets. Best scores are in **bold**.

| Method | B | M | R | C | S | CS | RCS |
|---|---|---|---|---|---|---|---|
| Flickr30K | | | | | | | |
| BLIP | 37.6 | 27.2 | 57.1 | 92.8 | 20.1 | 78.6 | 81.1 |
| +Remove | 38.6 | **27.4** | **57.5** | **95.8** | 21.0 | **79.2** | 81.9 |
| +ReplaceCap | 37.9 | **27.4** | 57.4 | 94.5 | **21.1** | 78.9 | 81.5 |
| +ReplaceImg | **39.0** | 27.3 | 57.4 | 95.7 | 20.7 | 79.1 | **82.0** |
| COCO | | | | | | | |
| BLIP | 39.9 | 30.8 | 59.9 | 132.0 | 23.8 | 77.3 | 82.8 |
| +Remove | 40.1 | 30.9 | 60.0 | 132.5 | 23.6 | 77.3 | 82.8 |
| +ReplaceCap | **40.2** | 30.9 | **60.1** | 132.7 | **23.9** | 77.3 | 82.8 |
| +ReplaceImg | **40.2** | **31.0** | **60.1** | **133.1** | **23.9** | 77.3 | 82.8 |

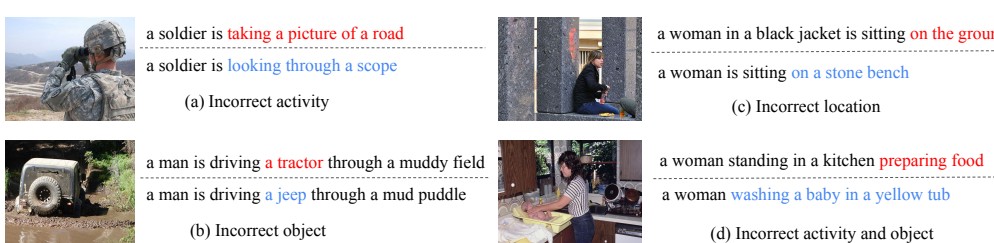

(a) Incorrect activity

a soldier is taking a picture of a road
a soldier is looking through a scope

(c) Incorrect location

a woman in a black jacket is sitting on the ground
a woman is sitting on a stone bench

(b) Incorrect object

a man is driving a tractor through a muddy field
a man is driving a jeep through a mud puddle

(d) Incorrect activity and object

a woman standing in a kitchen preparing food
a woman washing a baby in a yellow tub

Figure 4: Qualitative examples from the COCO dataset of captions generated by the BLIP model (top), and the same model trained using our REPLACEIMG data curation (bottom). The errors made by the BLIP model (shown in red) are avoided by REPLACEIMG curation (shown in blue).

## 4 Experiments

We evaluate our data curation methods on the MS COCO and Flickr30K datasets when finetuning the pretrained BLIP [28] model. We evaluate the captions using **B**LEU [37], **M**ETEOR [10], **R**OUGE [31], **C**IDEr [51], **S**PICE [2], **CLIPS**core, and **R**ef**CLIPS**core [18].

We use the ViT-based BLIP model [28] as our captioning model. We note that BLIP has a captioning and filtering (CapFilt) data augmentation process during its pretraining, where both components were finetuned on the COCO dataset. Therefore we use pretrained checkpoint $BLIP_{CapFilt}$ for Flickr30k and $BLIP_{base}$ for COCO in our experiment, removing the effects from the CapFilt process. We finetune BLIP using a batch size of 128 for 5 epochs on $4\times$ A100 GPUs.

### 4.1 Results

**Removal/Caption Replacement** As shown in Table 2, dynamically removing mismatched image-text pairs or replacing captions can effectively improve performance on both datasets over baselines on all metrics. For Flickr30K, the dynamic updates work best when apply to the top 1% of high-loss samples for REPLACECAP, and to samples whose loss are two standard deviations higher than the mean for REMOVE. For COCO, both REPLACECAP and REMOVE works best when curating the top 1% of high-loss samples. We repeat that during the curation process, no additional data samples or computation cost is introduced. We further study the effect of the amount of curation in Section 5.

**Image Generation-based Replacement** We evaluate Image Generation-based Replacement on both the Flickr30K and COCO dataset. During finetuning, we replace images in the original text-image pairs with Stable Diffusion-synthesized images (ReplaceImg in Table 2). The results show improvements compared to the baseline in every evaluation measure with best performance obtained at replacement ratio of 40% for Flickr30K and at 10% for COCO. We show qualitative examples in Figure 4, where models finetuned with our proposed curation method can generate better captions for some scenes that may confuse the standard finetuned model. In Section 5.1, we analyze the effects of varying the amount of synthetic images replaced, and in Section 5.2, we conduct a human study of the types of errors found in the generated images.

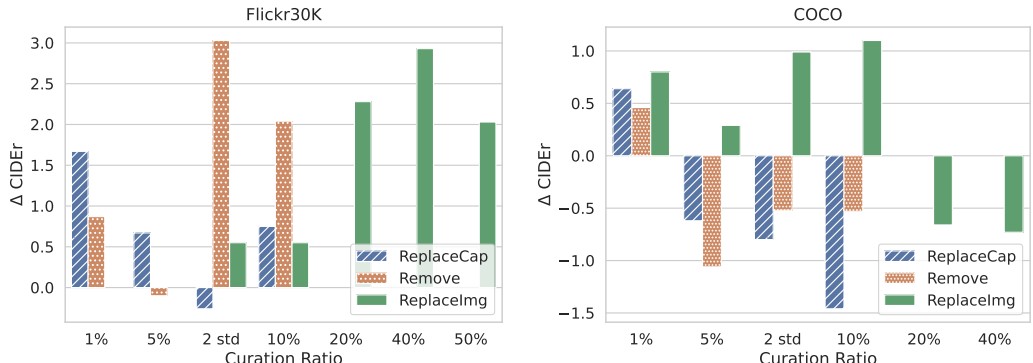

Figure 5: Effects of the amount of data curated when finetuning the captioning model. We can observe that Flickr30K needs more curation (40% REPLACEIMG or 2 std REMOVE) than COCO (10% REPLACEIMG or 1% REPLACECAP). Flickr30K benefits more from removing high-loss training samples, indicating the original dataset may be noisier than MS COCO. For the 2 std approach, the number of samples curated is not fixed after each epoch and varies between 5% to 10%.

# 5 Analysis and Discussion

## 5.1 Data Curation: how much and when?

We analyze how the amount of curation affects image captioning performance. We examine different ratios of training samples that are removed, replaced with an alternative caption, or replaced with a synthesized image. For REMOVE and REPLACECAP, we consider curation ratio of 1%, 5% and 10% of high-loss samples. For REPLACEIMG, we consider 10%–80% curation ratio. In addition to fixed X% ratios, we also intereven on samples that have losses two standard deviations worse than the mean.

**Flickr30K needs more curation than COCO.** The results of this analysis are shown in Figure 5. The best improvement in performance for Flickr30K is achieved either through removing high loss samples that are two standard deviations away, or replacing images for 40% of the high loss samples.

In the COCO dataset, replacing images for 10% of the high loss samples gives the best improvement compared to no data curation. The second best performing method for COCO is removing or replacing captions of only 1% of the high loss samples. This indicates that Flickr30K may contain more noisy samples than the MS COCO dataset. Compared to MS COCO, Flickr30K contains more samples with long captions (Figure 6), which may include overly-specific details that are inconsistent with other captions and are hard for the model to learn. See more examples in our supplemental materials. Through our curation-based finetuning, these samples can be effectively identified, removed or replaced, which indicates that

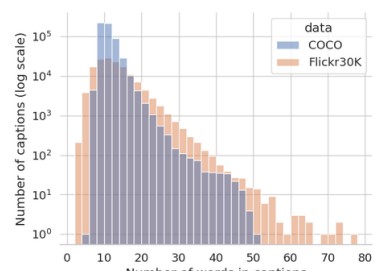

Figure 6: Distribution of caption lengths.

our method is efficient when training with noisy datasets. We note that curating more than 50% of the data does not benefit training and actually harms performance.

**Static image replacement versus dynamic replacement** In REPLACEIMG (Section 3.3), we dynamically replace images for the difficult training samples. Another static approach is to replace the identical images, i.e. $I_k$ in $\{(I_k, C_k^1), \ldots, (I_k, C_k^J)\}$, with unique SD-synthesized images before training, instead of updating the training samples while training. With static image replacement, for each of the reference captions, we replace their original image with a SD-synthesized image. Static replacement with 20%–80% curation ratio corresponds to replacing images for one–four captions of

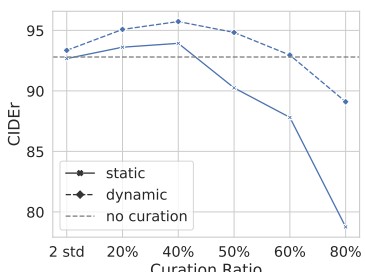

Figure 7: Dynamic image replacement against static replacement, as a function of the number of samples replaced.

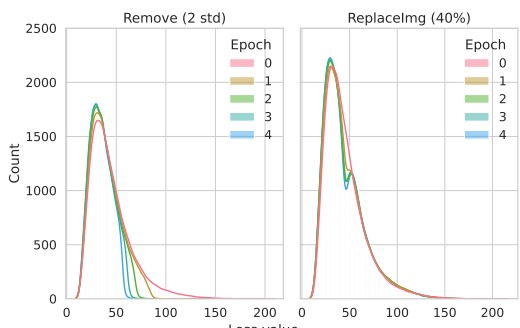

Figure 8: Loss distribution of training samples across epochs with different curation methods.

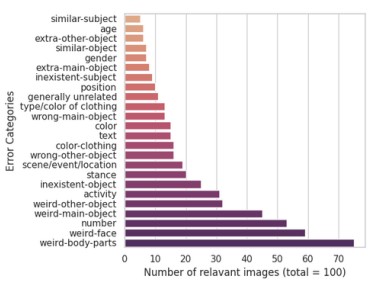

(a) Distribution of text-to-image generation errors.

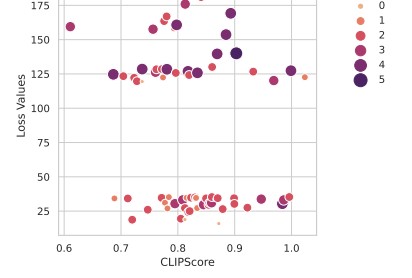

(b) Human evaluation versus CLIPScore.

Figure 9: Results of the human study of the errors made by the Stable Diffusion model in 100 images. The images used in the study were chosen to represent either low or high model loss. (a) Histogram of the number of errors annotated in each category. The most frequently occurring annotations concern weird deformations in the expected objects or humans. (b) Relationship between average number of identified errors by human annotations for each synthesized image and its captioning loss with regard to original captions. More errors are identified in images of higher loss. However, CLIPScore appears to fail in validating qualities of the synthesized images, as the score ranges are almost identical for samples that contain more errors.

the original five. The 50% replacement ratio mimics a fair coin-flip, where for each of the text-image samples, there is 50% probability for the image to be replaced by a synthesized image.

We compare the efficacy of these two approaches in Figure 7. When evaluating on the original 1k validation set, we see that for both approaches, incorporating synthesized images of 20% or 40% can assist finetuning and achieves higher BLEU4 and CIDEr scores. Nevertheless, dynamic image replacement consistently performs better than the static method, showing focusing on the hard samples is effective. For both replacement methods, performance starts to decrease when the curation ratio is too high. This may indicate that when incorporating too many images from the synthetic distribution, the gap increases between the training and evaluation sets.

Figure 8 shows the effect of the curation techniques in the training loss distributions across epochs. For the REMOVE approach, training samples with loss that are two standard deviations worse than the mean are dynamically removed during training, leading to the shrinking tail of the loss distribution. SD-based image replacement gradually reduces losses through learning from a mixture of Gaussian distribution from original image-text pairs and the ones contain synthesized images.

| Image | Caption | CLIPScore | Loss | Categorized Errors |
|-------|---------|-----------|------|---------------------|
| | A picture of two women with one in lacy white dress with handbag and leggings and the other with a tall red hat, black mid-dress, and frame like plastic dress on top. | 84.1 | 181.0 | type/color of clothing, color-clothing, weird-face |
| | A pedicab driver waiting on his bike. | 89.3 | 169.2 | weird-main-object, weird-other-object, weird-body-parts, stance |
| | A man in a black suit with tie and corsage smiles at a girl who smiles back, both are sitting at a table at a semi formal event such as a wedding or reunion. | 77.6 | 163.5 | color-clothing, weird-body-parts, wrong-main-object, scene/event/location |
| | Two men are playing guitars and one man is singing into a microphone on a stage with the spotlight on them. | 74.7 | 26.0 | weird-face, weird-body-parts, weird-main-object, weird-other-object |
| | There a several people in a dark bar-type room, including one girl on a stool. | 84.9 | 26.5 | number, weird-face, weird-main-object, weird-body-parts |
| | Many children are playing and swimming in the water. | 78.2 | 26.9 | weird-face, weird-body-parts |

Figure 10: Examples of synthesized images that are of high losses (top) and examples of synthesized images that are of low losses (bottom). Human annotations show that consistent error types have been recognized for the high loss samples while CLIPScore fails to align with human judgement. The low loss synthesized images are visually less complicated than the higher loss ones, but can still often look weird and contain errors in color or objects.

## 5.2 Human Study: Errors made by SD models

Finally, we conduct a human study of the errors present in the SD-synthesized images. This will serve to better understand any shortcomings with this approach that is not captured by automatic evaluation measures.

We first ranked SD-synthesized images by model loss from the 1K images in the validation set. This validation set of synthesized images was generated using the best performing configuration of the Stable Diffusion model (see Section 3.3). We then sampled a subset for human annotation using the top and bottom 50 images based on their loss using our fine-tuned captioning model. These images are uniformly divided into 5 sets, each containing 20 images with equal number of the high loss ones and the low loss ones. The data was annotated by 12 people, members of a university research lab with a basic understanding of Stable Diffusion but no knowledge of the bi-modal distribution of images. The annotators were asked to categorize the errors they observed in the synthesized images, given both the image and the reference sentences that were used to generate the images. Each participant annotated one set of 20 images.

Starting from the categories defined by van Miltenburg and Elliott [50], we predefined 25 categories including general errors such as color, or number mismatches, and errors related to people and objects in the images. Please see the user interface in supplemental materials. We analyze the human judgements for the images that have at least three annotations, yielding 74 unique images.

As shown in Figure 9a, the most common problem of SD-synthesized images are that they often generate weird face or body parts, which makes the images less natural or pleasant. The Stable Diffusion model is also weak at generating the correct number of people or objects. From Figure 9b we confirm the quality of our collected annotations that high loss figures often contain more errors on average. Furthermore, we note that CLIPScore does not appear to align with human judgements, indicating its weak capability of evaluating quality of generated images. Please see more concrete examples in Figure 10.

## 6 Conclusion

In this paper, we have shown a simple, yet effective, data curation framework that can improve the performance of image captioning models. We investigated three approaches to data curation that dynamically update the training dataset based on high-loss image-caption samples. The methods involved either removing a sample, replacing the caption in a sample, or generating a new image from existing captions. Experimental results on the Flickr30K and MS COCO datasets show the effectiveness of these approaches to data curation without increasing the total size of the training dataset. A deeper analysis of the images synthesized by Stable Diffusion shows frequent errors on generating objects of a certain amount or color, and struggles with human body features. A human evaluation of the errors in those images shows a clear difference in images with high or low losses.

In the future, we expect that better text-to-image generation models will lead to further improvements from using synthesized images for difficult captions in existing training datasets. We plan on verifying whether these findings extend to other image captioning models, which was not possible here due to computational issues. Finally, we are interested in applying the same framework to other multimodal tasks, especially those with undercomplete datasets that cannot comprehensively cover the distributional space due to the cost of crowdsourcing enough data, e.g. visual question answering, or visually-grounded dialog.

## Limitations

While our curation methods being effective on image-captioning in the finetuning and fewshot-learning settings, it is not clear if the same strategy can be scaled and adapted also to vision-language pretraining. Currently our data curation methods also rely on state-of-the art pretrained models for both image understanding and text-to-image generation. In pretraining, models will often be trained from scratch and pretraining data are often collected from multiple datasets and resources.

Moreover, while we take an online approach to data curation, our current approach is upper bounded in speed and performance of the text-to-image generation model. This might be a large bottle neck for adapting the strategy for more complicated vision-and-language tasks.

## Ethics Statement

Text-to-image generation with Stable Diffusion is controversial in the broader AI and ethics community[6]. For example, it can generate images according to gender or racial stereotypes, which may prove harmful to members of those communities [30]. In this paper, we use Stable Diffusion to improve the quality of an image captioning model, given a specific set of crowdsourced captions. Those captions may themselves contain harmful stereotypes that would become more prevalent in our dynamically updated training datasets. As we dynamically update the model with new images based on loss values, we remove the water-marker in our generated images to prevent information leak to the model. Use of the synthesized images will strictly follow community guidelines.

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
