# OpenReview forum: "Data Curation for Image Captioning with Text-to-Image Generative Models"
_NeurIPS.cc/2023/Conference — Submitted to NeurIPS 2023_

### Official Review · Reviewer_tP6L · 2023-06-15

**Soundness:** 2 fair
**Presentation:** 3 good
**Contribution:** 2 fair
**Rating:** 4
**Confidence:** 4

**Summary:**

This paper focuses on data curation for image captioning. This paper shows that mismatched image-caption pairs do harm to the captioning model. To address this problem, generative models are used. In detail, the BLIP model is used to generate captions based on images, and the Stable Diffusion model is used to create images based on captions.

**Strengths:**

1. The data curation is an important and effective topic, which could benefit many tasks including visual synthesis, image captioning, language and visual representation, etc.
2. This paper discovers the weak point of captioning datasets, especially for the Flicker30K.
3. It is interesting to use the BLIP model and the Stable Diffusion model to create data for training.

**Weaknesses:**

1. There are many methods to augment text dates, e.g., adding or editing some words, using synonyms, and changing sentence structure. I think these methods are also worth evaluating.
2. From Table 2, we can see that the BLIP's performance is not significantly affected by the methods proposed in this paper (i.e., Remove, ReplaceCap, and ReplaceImg). For example, the CIDEr of COCO only slightly raises from 132.0 to 133.1.
3. Figure 5 shows that the proposed methods might make the performance worse, especially in the COCO dataset. So I am concerned about the generalization of the proposed methods.

**Questions:**

Both the BLIP model and the Stable Diffusion model use a large size of datasets (e.g., LAION-5B). Would the performance of Image Captioning be improved by using part of the data in LAION-5B?

**Limitations:**

None.

---

> ### Author Rebuttal · Authors · 2023-08-09
>
> > Evaluation of text augmentation methods
>
> Yes there are many possible text augmentation methods, which mainly involve text augmentation which increases the total number training samples, such as [[3]](https://www.mdpi.com/2076-3417/10/17/5978). Instead, we focus on the new approach that leverages the text-to-image models to dynamically curate existing images without scaling training data.
>
> ---
> > Would the performance of Image Captioning be improved by using part of the data in LAION-5B?
>
> Yes, we believe that this is possible with some filtering mechanism, as BLIP gained its performance improvement with the CapFilt procedure during pretraining. We are not sure if the finetuning performance can be further improved with the data already seen in the pretraining stage (though BLIP only uses part of LAION-5B).
>
> ---
> > BLIP's performance is not significantly affected by the methods proposed. For example, the CIDEr of COCO only slightly raises from 132.0 to 133.1.
>
> Finetuned with our curation methods, BLIP reaches 95.8 CIDEr score with a +3 CIDEr score increase, which is a state-of-the-art performance on Flickr30K image captioning. Though the absolute improvement of CIDEr for COCO may seem small, this is already comparable to the BLIP_CapFilt model (133.3 CIDEr), whereas the CapFilt technique is applied during pretraining and would require more computation resource.
>
> ---
> > generalization of the proposed methods
>
> Please kindly see our general response for the generalizaiton ability of our approach.

---

> > ### Comment · Reviewer_tP6L · 2023-08-14
> >
> > Thanks for the author's elaborate response, and most of my concerns have been well addressed.

---

> > > ### Author Response · Authors · 2023-08-14
> > >
> > > Thank you! We are glad to hear that our response did a good job of addressing most of your concerns. Does it change your final rating for our submission?

---

### Official Review · Reviewer_9L7M · 2023-07-05

**Soundness:** 2 fair
**Presentation:** 2 fair
**Contribution:** 2 fair
**Rating:** 3
**Confidence:** 4

**Summary:**

This paper studies data curation strategies for training image captioning models. Firstly, it identifies the “difficult samples” based on the captioning loss dynamically at the end of each epoch. Subsequently, it introduces three data curation strategies to modify the difficult samples: (1) removal of an image-text pair, (2) replacement of the caption and (3) replacement of the image using text-to-image generative models. The main technical innovation is the third strategy, which is carefully designed in terms of prompt engineering and fine-tuning on the image captioning datasets.

The empirical studies show that the proposed data curation strategies can enhance the performance of the baseline BLIP captioning model. The authors also conduct analysis on the data curation ratio, dynamic versus static curation strategy and the errors of images generated by the stable diffusion model.


**Strengths:**

* The idea of employing text-to-image generative models to curate training data for image captioning is novel and well-motivated.

**Weaknesses:**

### Effectiveness of the proposal
* According to Table 2, the performance of the third data curation strategy, which is the main technical innovation of this work, is not advantageous compared to the heuristic removal and caption replacement strategies.
* According to Figure 5, all three proposed strategies are sensitive to data curation ratio. Consequently, training the captioning model multiple times is necessary to achieve satisfactory performance, which is less efficient compared to the baseline BLIP model.

### Design of the method
* Identifying the samples to modify based on training loss is questionable. A higher loss does not necessarily imply that the sample is harmful to training. Although Section 5.2 has shown that more errors are identified in images of higher loss, the experimental setup has two issues: (1) The loss is computed over the generated images rather than the real images in the original dataset. (2) The errors are categorized as targeting image generation, rather than image captioning. In other words, an image that possesses imperfect visual quality but aligns well with the caption may not necessarily be considered a noisy training sample for image captioning.

### Missing reference
* An idea similar to the “round-trip captioning evaluation” is already proposed by [1], which generates a caption from the synthesized image and measures the similarity between input text and predicted caption.

### Clarity
* In line 243, it is unclear whether the “model loss” refers to captioning loss or image generation loss.

[1] Inferring Semantic Layout for Hierarchical Text-to-Image Synthesis.


**Questions:**

N/A

**Limitations:**

The authors have discussed the limitations of this work from three aspects: (1) Lack of adaptation of the proposal to the pre-training stage. (2) Reliance on pre-trained image understanding and text-to-video generative models. (3) Increase in training time due to the usage of text-to-image generative model. Moreover, a significant limitation of this study is the absence of evidence demonstrating the superior effectiveness of the generated images compared to heuristic data curation strategies.

---

> ### Author Rebuttal · Authors · 2023-08-09
>
> > Effectiveness of the proposed ReplaceImg method
>
> In Figure 5, we show that ReplaceImg generally works better for both datasets (second best for Flickr30k---0.1 CIDEr lower than REMOVE, and best for COCO). Flickr30k benefits more from removing high-loss training samples indicates the original dataset maybe noisier than COCO (Figure 6 and L211). This is more specific to the dataset instead of the general image captioning task or our curation method.
>
> ---
> > Limitation: lack of adapting the approach to pretraining
>
> We agree with the reviewer that our approach could be adapted to pretraining vision-language models to learn better general visual representations, and more applications. However, that would require substantially more computational resources than the present paper.
>
> ---
> >  sensitivity to data curation ratio: Do we need to train the model for multiple times for the curation method to work?
>
> We conducted additional experiments to show that the curation approach is model-agnostic and the ratio is transferable. We evaluated our curation methods on another state-of-the-art VL model---BEiT-3 by applying exactly the same curation ratio from BLIP and obtained similar improvements. Please kindly see our general response and the attached PDF for detailed experiment results.
>
> ---
> > Identifying the samples to modify based on training loss is questionable. A higher loss does not necessarily imply that the sample is harmful to training.
>
> The use of loss values to separate difficult samples have been discussed and utilized in previous literatures including Curriculum Learning and Self-paced Learning. Please kindly see our general response for more details. And to clarify, we didn't use the training loss, as in L82-86, we use the model checkpoint after each epoch to evaluate on the training samples and use the loss as the indicator of the sample difficulty.
>
> ---
> > Regarding section 5.2: (1) The loss is computed over the generated images rather than the real images in the original dataset. (2) The errors are categorized as targeting image generation, rather than image captioning.
>
> 1) To clarify, Secontion 5.2 serves to point out limitations in the synthesized images (L240) instead of the issues in the original dataset. Please refer to the examples of the high loss samples in the original training dataset in the Appendix Figure 2. Our ReplaceImg approach replace images of the high loss samples dynamically regardless of whether the image belongs to the original dataset. A low-quality synthesized image of high loss will also be replaced in the following training epoch.
>
> 2) The errors in the synthesized images are identified by human annotators given reference captions. Please see our annotation interface in Appendix Figure 1. During the study, the annotators are asked to judge the quality of the synthesized image in the context of whether the image has issues to match the reference captions.
>
> ----
> > Clarity: model loss in L243
>
> The model loss refer to the captioning loss in L243, we will make it more clear in the revised version.
>
> We thank the reviewer for pointing out the missed reference, and we will make sure to update our revised version.

---

> > ### Comment · Reviewer_9L7M · 2023-08-17
> > **Response to Author Rebuttal**
> >
> > Thank you for taking the time to respond to my comments. However, my major concerns still remain and I prefer to keep the rating unchanged.
> >
> > ### Effectiveness of the proposed ReplaceImg method
> > While ReplaceImg performs the best in COCO and the second best in Flickr30K, the advantage over the heuristic data curation strategies is marginal (-0.1 CIDEr score than Remove in Flickr30K, +0.4 CIDEr score than ReplaceCap in COCO).
> > ### Sensitivity to data curation ratio
> > * The additional results of BEiT-3 are still insufficient to demonstrate the curation ratio is generalizable to different VL models.
> > * As for the cross-domain evaluation, the additional results show that the best curation ratio in COCO (10%) fails to improve over the baseline BLIP and BEiT3 in the COCO -> Flickr30K setting.

---

> > > ### Author Response · Authors · 2023-08-18
> > >
> > > We thank the reviewer for reading through our response and providing the feedback!
> > >
> > > > The additional results of BEiT-3 are still insufficient to demonstrate the curation ratio is generalizable to different VL models.
> > >
> > > Can you explain why showing that it directly translates to another model is insufficient?
> > >
> > >
> > >
> > > > Cross-domain evaluation
> > >
> > > We think the reviewer's ask for improved cross-domain performance would be more relevant to pretrained models instead of the finetuned models we present in the paper. As widely known as the impossible triangle [1], finetuned models often struggle with OOD generalization [2]. Nevertheless, we provide cross-domain evaluation results as we consider it as an interesting question. And we are glad to see that our curation method does not hurt cross-domain performance (COCO-> Flickr30K), and has gained significant improvements (+9.6 CIDEr score increase) compared to the standard finetuned model when transferring from Flickr30K to COCO. This already shows the model finetuned with our curation method has good enough cross-domain performance.
> > >
> > > [1] Zhu, C., & Zeng, M. 2022. Impossible Triangle: What's Next for Pre-trained Language Models?
> > >
> > > [2] Aishwarya Agrawal, Ivana Kajic, Emanuele Bugliarello, Elnaz Davoodi, Anita Gergely, Phil Blunsom, and Aida Nematzadeh. 2023. Reassessing Evaluation Practices in Visual Question Answering: A Case Study on Out-of-Distribution Generalization

---

### Official Review · Reviewer_ymVw · 2023-07-06

**Soundness:** 1 poor
**Presentation:** 2 fair
**Contribution:** 1 poor
**Rating:** 3
**Confidence:** 5

**Summary:**

This paper proposes a data curation model for image captioning. If the loss of a particular image caption pair is high, then either remove the image-caption pair from the training set or replace the caption with a more similar caption or they generate a new image for the difficult caption. The authors demonstrate these strategies help to improve the performance of BLIP caption generation model.


**Strengths:**

The idea is interesting.

Paper shows some positive gains on COCO and FLickr30K.


**Weaknesses:**

The details of the method are not clear. How to select a replacement caption?

Why one should pick only the high-loss image-caption pairs? Loss may be high due to many other reasons.

Compared to other data augmentation methods in the literature that is also discussed in the related work, what is the novelty?

Why this is a significant finding? I am not sure if this is a significant finding.

The method is also evaluated using a single model.

Obtained results are not state-of-the-art.

It is not clear whether such a mechanism will contribute to any state-of-the-art methods in captioning.

**Questions:**

What are the conceptual differences w.r.t. [3]? W.r.t [3] is this paper novel?

I am not able to understand Figure 2 and the message behind this figure.

No guarantee synthesized image has a smaller loss than the original image.

It is not clear to me what is F in Table 1.


**Limitations:**

Limitations and societal impact are discussed in the paper.

---

> ### Author Rebuttal · Authors · 2023-08-09
>
> > How to select a replacement caption?
>
> This is described on  L98-102. As for both Flickr30K and COCO, each image is paired with 5 caption annotations. We replace the caption by randomly selecting from the other 4 captions.
>
> ---
> > Significance of our findings
>
> We propose a model-agnostic approach to dynamically updating an image captioning dataset while it is being used to train an image captioning model. With the proposed method, we show state-of-the-art image captioning models can be improved by curating on existing resources.
>
> ---
>
> > The method is evaluated using a single model
>
> We conduct additional experiments to evaluate our curation methods on another state-of-the-art VL model BEiT-3. We used exactly the same replacement rates from the BLIP model. The results show that we obtained a similar improvements in performance  by directly applying those replacement rates, i.e. 3 CIDEr points improvements on Flickr30k with ReplaceImg (40%) and 0.7 CIDEr points on COCO with ReplaceImg (2std). The curation is more effective on Flickr30K, which may be because COCO is included in the BEiT-3 pretraining data. Please see the detailed results in Table 1 in the rebuttal PDF.
>
> | BEiT-3              |       B4 |     CIDEr |
> |---------------------|---------:|----------:|
> | Flickr30K           |     28.9 |      79.3 |
> | +ReplaceImg (40%)   | **32.0** |  **82.4** |
> | COCO                |     39.4 |     133.7 |
> | +ReplaceImg (2 std) | **39.6** | **134.4** |
>
> ---
>
> > Difference to [3]
>
> There are several differences. First, [3] proposes to perform data augmentation to the captions, whereas we perform data augmentation of the images. Second, [3] performs the data augmentation as a pre-processing step, whereas our data curation happens dynamically as the model is trained. Third, [3] reports their final results after additional fine-tuning with SCST whereas we only use cross-entropy based training of the model. Finally, [3] augmented the training dataset to 2-3 times larger by augmenting the image captions, whereas we show improved performance by curating on the existing dataset without increasing the total number of unique training examples. In other words, it is an in-place augmentation.
>
> ---
>
> > Figure 2
>
> This is covered in  Section 3.1 and especially L91-94. Together with Figure 8, we analyze how the curation methods impact the loss distribution of training samples, and help with the model training process.
>
> ---
>
> > No guarantee synthesized image has a smaller loss than the original image.
>
> There is indeed no guarantee that the synthesized image will result in improving the quality of the model. . It is possible that the synthesized image has a larger loss than the original image. However, as we dynamically curate the dataset during the training process (L38 and Figure 7),  the synthesized images/captions that have high losses would be replaced in the following training process if they prove to be be in the tail of the distribution of training losses.
>
> ---
>
> > F in Table 1
>
> F in Table 1 indicates that we finetune the stable diffusion model. Thanks for pointing out the issue, we will make it more clear in the revised version.
>
> ---
>
> > Why one should pick only the high-loss image-caption pairs?
>
> We use the loss values to identify the difficult samples during training (Section 3.1). The use of loss values to separate difficult samples have been discussed and utilized in previous literatures including Curriculum Learning [1] and Self-paced Learning [2]. Please see the general response for more details on the effectiveness of dynamic loss-value based curation.
>
> [1] Yoshua Bengio, Jerome Louradour, Ronan Collobert, and Jason Weston. Curriculum learning. In Proceedings of the 26th annual international conference on machine learning, pp. 41–48. ACM, 2009.
>
> [2] M Pawan Kumar, Benjamin Packer, and Daphne Koller. Self-paced learning for latent variable models. In Advances in Neural Information Processing Systems, pp. 1189–1197, 2010.

---

> > ### Comment · Senior_Area_Chairs · 2023-08-19
> > **Please discuss**
> >
> > Dear ymVw,
> > Thanks for your review! Are your main concerns addressed by the rebuttal? Also, you explicitly mentioned that the results are not state-of-the-art. Could you please provide explicit references to methods that perform better?
> >
> > Best,
> > SAC

---

> > ### Comment · Reviewer_ymVw · 2023-08-22
> >
> > Thanks for the response. I am not still convinced that the use of loss value is the right approach. Loss is an effect of many causal elements. After reading other reviews and all responses, I will keep my original rating.

---

### Official Review · Reviewer_Hsed · 2023-07-07

**Soundness:** 3 good
**Presentation:** 3 good
**Contribution:** 2 fair
**Rating:** 4
**Confidence:** 4

**Summary:**

This paper focuses on improving image captioning by improving the quality of the existing dataset. To this end, this paper proposes three data curation methods: the removal of an image–caption sample; replacing a caption with another caption; and replacing images using a text-to-image generation model. Experimental results demonstrate that models trained with the proposed methods consistently outperform baselines.

**Strengths:**

1.The proposed method is well-motivated, that is to improve the quality of the existing dataset. This paper explores the problem of making better use of existing datasets, which is a very interesting research direction.

2.The authors conduct extensive experiments over these two datasets, where the models trained with the proposed methods outperform baseline methods consistently.

3.The paper is well-written and easy to follow.

**Weaknesses:**

1.From my understanding, it is risky to judge the quality of the sample based on the loss value. A sample with a large loss value may be a hard sample or a mislabeled sample, so it is risky to judge the sample quality only based on the loss value.

2.In addition to the performance of the model on the test set, the generalization ability of the model is also important. It is not clear whether the proposed method reduces the gap between the training set and the test set or improves the quality of the training set.

3.Lack of necessary theoretical analysis.

**Questions:**

Can the proposed method improve the generalization ability of the model? It would be better to carry out cross-domain testing to verify the generalization ability of the model, that is, the Flickr30K dataset is used as the training set, and the test set of the COCO dataset is used as the test set.

**Limitations:**

yes

---

> ### Author Rebuttal · Authors · 2023-08-09
>
> > Generalization ability and cross domain evaluation
>
> It had never occurred to us that our data curation method would reduce the gap between the training and the test set because nothing in the method knows anything about the distribution of the test data. In order to better understand how our method contributes to generalization, we adopt your suggestion and conduct two additional experiments. Please kindly see the cross-domain evaluation results and more details about strong generalization ability of our approach in the general response.
>
> ---
> > It is risky to judge the quality of the sample based on the loss value
>
> Please kindly see the general response for this question.
>
> ---
> > Lack of necessary theoretical analysis
>
> We are not sure how to interpret this comment. Which type of theoretical analysis do you want to see in the paper?

---

> > ### Comment · Reviewer_Hsed · 2023-08-21
> >
> > Thanks for the author's response. In general, theoretical analysis refer to why and how the proposed method works.

---

### Official Review · Reviewer_ndRN · 2023-07-08

**Soundness:** 2 fair
**Presentation:** 4 excellent
**Contribution:** 2 fair
**Rating:** 3
**Confidence:** 5

**Summary:**

In this paper, the authors propose an iterative training approach to improve image captioning models.
This approach _refreshes_ the training dataset every epoch with _higher quality_ image-text pairs (authors call it "data curation").
Dataset samples with very high training loss are updated -- the real image is replaced with one generated by the Stable Diffusion model.
The authors compare their approach with two baselines: one which removes high-loss samples, and one where the image is replaced by another from the training dataset itself.
Experiments are performed with the BLIP model and two captioning datasets -- COCO and Flickr30K.
Authors also perform an accompanying human study to provide directions for future work.

**Strengths:**

This paper has numerous technical strengths:

- The proposed method is conceptually simple and easy to implement.
- The strategy of updating the training dataset with "better" samples is very general: it is agnostic to the model architecture and the multi-modal task at hand.
- The writing and presentation quality of the paper is excellent. It contains adequate implementation details to make this work reproducible.
- The experimental setup and ablation study is very meticulous. Tables of results contain experiments that begin with a BLIP baseline, and subsequent rows introduce one change at a time.
- The authors have conducted a human study with sensibly defined failure categories to understand how failure modes of Stable Diffusion can impact captioning performance.


**Weaknesses:**

Like its technical strengths, this paper also has some shortcomings.
Below I list a few salient concerns with the paper.
I look forward to hearing the authors' response, and I am happy to update my final assessment.

1. **Results do not match with the presented story:**
The main results (`Table 2`) indicate that all considered dataset curation approaches are beneficial over a BLIP baseline that doesn't train on curated data.
However, the main pitch of this paper is to use generative models like Stable Diffusion to replace images (last row),
which in fact performs marginally better or even worse than other curation techniques.
The biggest improvements are generally yielded by "Remove" strategy.
I recommend the authors rethink the positioning of motivation and frame it as an exploratory study --
it seems obvious to use generative models for iterative training/distillation and some works already do it for other applications,
but for this task, a practitioner is better off by simply filtering noisy samples altogether.

2. **Captioning metrics appear saturated, maybe overkill for COCO/Flickr:**
The captioning metrics on COCO and Flickr are already saturated,
e.g. decimal improvements are less meaningful for COCO in the range of 130+ CIDEr and 20+ SPICE score.
Since BLIP is already rained with large amounts of data and diverse tasks,
the proposed approach may be an overkill for the tasks considered in this paper.
I suggest the authors rethink other applications where the benefits of this strategy are more prominently observed (see Weakness 5 below).

3. **What if the caption is noisy and can't generate meaningful images?**
An image-text pair may be unaligned if the caption is uninformative,
as frequently encountered in larger web datasets like
[Conceptual Captions](https://arxiv.org/abs/2102.08981),
[YFCC](https://arxiv.org/abs/1503.01817),
[RedCaps](https://arxiv.org/abs/2111.11431), etc.
For instance, captions coming from alt-text may not have any semantic content whatsoever
(e.g. see Figure 2 in [ALIGN paper](https://arxiv.org/abs/2102.05918)) to generate meaningful images.
The proposed approach forces the generative model to create an arbitrary image and ends up adding noise to the training data.
Some selective mechanisms to replace either image or caption may be needed to scale this approach to general image captioning beyond COCO and Flickr30K.

4. **Related work needs more coverage:**
The main focus of this paper is image captioning, hence a broad coverage of prior works on image captioning is necessary.
However, this section only cites a handful of very recent modeling papers.
I suggest the authors begin the discussion with some early image captioning papers like:

  - (Vinyals et al, CVPR 2015) Show and tell: A neural image caption generator
  - (Karpathy and Li, CVPR 2015) Deep visual-semantic alignments for generating image descriptions
  - (Donahue et al, CVPR 2015) Long-term recurrent convolutional networks for visual recognition and description

5. **[Related to 1, 2] Have the authors considered applications others than image captioning?**
What if this curation strategy is used to train general visual representations?
I suggest a CLIP-style contrastive model and/or BLIP/VirTex-style generative model.
The contribution can be strengthened by broadening the scope to various downstream tasks.


**Questions:**

Some comments and suggestions:

- Imagen is cited twice (44 and 45). Please remove the duplicate.
- Related work section has phrases like "large-scale stable diffusion models" (`Line 64`) and "stable diffusion text-to-image models" (`Line 69`).
  "Stable Diffusion" is a set of models developed by Stability AI startup, and these phrases seem to appropriate a brand as a mathematical/conceptual term.
  I suggest the authors remove "stable" from these phrases and call them "text-to-image diffusion/generative models" or something.


**Limitations:**

The limitations section should be updated if any of the above-mentioned open questions are not within the scope of this paper.

---

> ### Author Rebuttal · Authors · 2023-08-09
>
> We thank the reviewer for the helpful feedback and recognising the contribution of our work and its potential impact to the community.
>
> > Results do not match with the presented story: simple removal works better than ReplaceImg.
>
> Figure 5 gives a broader context than Table 2, where it can be seen that ReplaceImg yields more reliable improvements for the COCO dataset. Flickr30k marginally benefits more from removing high-loss training samples, which may indicate that the Flickr30K dataset is noisier than COCO (Figure 6 and L211). This is more specific to the dataset instead of the general image captioning task. For example, if we measure the overlap of tokens in the training and validation sets, an admittedly crude measure of similarity, we observe that, for Flickr30k, REMOVE improves the vocabulary similarity between train and test datasets by 15%, while the similarity improves 6% for COCO.
>
> > Captioning metrics appear saturated, maybe overkill for COCO/Flickr, rethink applications including learning general visual
> representations, etc. where the benefits of this strategy are more prominently observed
>
> We agree with the reviewer that our approach can be adapted to pretraining vision-language models to learn better general visual representations, and more applications, when computation budgets allow (L276-280). This is an exciting direction for future work that we will highlight in the paper. In the submitted paper, with our computational resources, we show that our proposed curation methods are beneficial and can be built upon existing state-of-the-art models.
>
> > What if the caption is noisy and can't generate meaningful images?
>
> Interesting question! As you wrote, this becomes more relevant if you were to apply the ReplaceImg method to larger-scale noisy datasets for pretraining. For our experiments on the Flickr30K and COCO datasets, we improved the expected generated image relevance by selecting the best prompting method from our round-trip captioning evaluation(L162-166). We reduce the impact of a noisy caption by concatenating all five captions and adding the styler. Please see Figure 1 in the rebuttal pdf for qualitative examples on the generated images. If we were to apply this to pretraining, it might be necessary to have an additional classifier that could determine whether a sentence was visually descriptive [[1]](https://aclanthology.org/W15-2805.pdf) or had a high expected semantic content.
>
> > Related work and phrasing suggestions
>
> We thank the reviewer for the detailed suggestions, and we will make sure to update the corresponding sections in the revised version.
>
> [1] Robert Gaizauskas, Josiah Wang, and Arnau Ramisa. 2015. Defining Visually Descriptive Language. In Proceedings of the Fourth Workshop on Vision and Language, pages 10–17, Lisbon, Portugal. Association for Computational Linguistics.

---

> > ### Comment · Reviewer_ndRN · 2023-08-21
> > **Response to author rebuttal**
> >
> > I thank the authors for a thoughtful rebuttal. I listed several concerns in my initial review, grouped into five bullet points. Below I provide pointwise responses outlining how far these concerns were addressed in the rebuttal:
> >
> > 1. **Results do not match with the presented story:**
> >
> > I am not entirely convinced by the response. This concern was raised by multiple reviewers, and (in my response) the authors restate their results, specifically `Figure 5` in the paper. I understand what the authors are trying to convey, but I was trying to make a broader commentary about the positioning of this paper. The authors present the story as "our curation method improves captioning performance", but the empirical results fall short to support this story — their approach works just as well as a baseline that removes noisy samples.
> >
> > In the past, the typical "minimum publishable unit" that advances SOTA on captioning benchmarks has had more gains (on CIDEr/SPICE metrics) than in this paper. For reference, authors can see a few papers on [Papers with code](https://paperswithcode.com/sota/image-captioning-on-coco-captions) and discount concurrent works published in the same conference venues. So in my opinion, the authors are competing on a losing ground and minimizing the impact of their work if they write "our curation method improves captioning" with the presented results. I suggested rethinking the positioning, to present a more exploratory study — it seems very obvious to readers that expanding the training data using diffusion models may be a "free lunch", but performance improvements are marginal in reality. Such a story may be an important piece of evidence that helps ML practitioners to calibrate their expectations on what these generative models can and cannot do. In fact, the human study presented in this paper would align well with an exploratory story. However, the authors do not comment on this suggestion.
> >
> > Below are recent examples in vision-language learning that authors may refer to consider how the presented results can be conveyed differently: (a) "The Curse of Recursion: Training on Generated Data Makes Models Forget" https://arxiv.org/abs/2305.1749 and (b) "Masked Autoencoding Does Not Help Natural Language Supervision at Scale" https://arxiv.org/abs/2301.07836
> >
> > 2. **Captioning metrics appear saturated, maybe overkill for COCO/Flickr, and (weakness 5) Have the authors considered applications other than image captioning:**
> >
> > The authors do not acknowledge the first half of my concern (+0.4 CIDEr on COCO is less meaningful when metrics are so saturated). They state limited computational resources as a reason to not explore other benchmarks and vision-language pretaining tasks. This argument is not convincing — the authors claim to use 4x A100 GPUs for experiments in the paper. In my opinion, this scale is sufficient to perform controlled comparisons with small-scale generative and contrastive vision-language models (e.g. CLIP with a ResNet-50/ViT-base and up to 4K batch size, or BLIP/VirTex with similar model capacity/batch size). Limited computational resources are a valid reason to discount the lack of additional experiments during the rebuttal. However, my concern broadly encompasses all the experiments presented in the paper. This could be a second alternative apart from framing this paper as an exploratory study — the authors should consider searching for downstream applications where their curation strategy can show more prominent improvements.
> >
> >
> > **(3) What if the caption is noisy and can't generate meaningful images? and (4) Related work needs more coverage:**
> > These concerns are sufficiently addressed, thank you.
> >
> > **Summary:** All things are taken together, this paper requires careful thought on (a) either presenting existing results in a less biased and more analytical manner or (b) demonstrating downstream tasks where the proposed approach is significantly better than baselines. In my opinion, saying that the proposed curation strategy is "effective" for improving image captioning models is not entirely correct. Unfortunately I will have to keep my original rating unchanged. I encourage the authors to update the paper and consider a future venue. Good luck!

---

### Author Rebuttal · Authors · 2023-08-09

We express our gratitude to all the reviewers for their time and helpful feedback. We are glad that all five reviewers found our work interesting and well-motivated, and `Reviewer-ndRn`, `Reviewer-Hsed` and `Reviewer-tP6L` also found our work enlightening to a broader scope of Vision-Language learning and useful for the community.

We address two shared concerns below.

> Identifying the samples as target of curation based on loss values (`Reviewer-Hsed`, `Reviewer-9L7M`)

The use of loss values to prevent difficult samples from confusing the model have been discussed and utilized in previous literature including Curriculum Learning [1] and Self-paced Learning [2]. Though we agree that there might be more advanced influence evaluations to judge the quality of the training samples, our experiments show that by curating on the samples that have outlier losses is sufficient to improve the downstream performance. High loss indicates the model is predicting the wrong probability distributions over the expected tokens given the image, i.e. struggles to generate a caption that is similar to the reference.

In Figure 7, we show the empirical effectiveness of replacing the images based on the loss criterion dynamically, instead of  randomly replacing the same amount of images. It is clear that the dynamic replacement of examples based on tracking the losses is always better than randomly replacing images. (The dashed-diamond line is always higher than the solid-circle line.) Qualitative examples in Figure 2 in the appendix also validates the effectiveness as the high-loss samples often have an overly specific caption that would be difficult for the model to learn/generate.

[1] Yoshua Bengio, Jerome Louradour, Ronan Collobert, and Jason Weston. Curriculum learning. In Proceedings of the 26th annual international conference on machine learning, pp. 41–48. ACM, 2009.

[2] M Pawan Kumar, Benjamin Packer, and Daphne Koller. Self-paced learning for latent variable models. In Advances in Neural Information Processing Systems, pp. 1189–1197, 2010.

---
> Generalization ability of the curation approach (`Reviewer-Hsed`)

In order to better understand how our method contributes to generalization, we conduct two different experiments.

First, we find that our curation approach is **generalizable to different VL models**. We evaluated our curation methods on another state-of-the-art VL model---BEiT-3 to see if our approach transfers. More specifically, we used exactly the same replacement rates from the BLIP model. The results show that we obtained a similar improvements in performance  by directly applying those replacement rates, i.e. 3 CIDEr points improvements on Flickr30k with ReplaceImg (40%) and 0.7 CIDEr points on COCO with ReplaceImg (2std). The curation is more effective on Flickr30K, which may be because COCO is included in the BEiT-3 pretraining data.

| BEiT-3              |       B4 |     CIDEr |
|---------------------|---------:|----------:|
| Flickr30K           |     28.9 |      79.3 |
| +ReplaceImg (40%)   | **32.0** |  **82.4** |
| COCO                |     39.4 |     133.7 |
| +ReplaceImg (2 std) | **39.6** | **134.4** |

Second, we conduct a **cross-domain evaluation** to determine if a model finetuned with our curation method has a stronger cross-domain generalization ability. We use the best performing BLIP model finetuned on Flickr30K (40% ReplaceImg curation), and evaluate on the COCO. We obtained +0.5 points BLEU score increase and a +2 CIDEr score increase compared to the standard finetuned model (no curation) on the COCO test set. For the BEiT-3 model, We obtained +3 points BLEU score increase and a +9.6 CIDEr score increase. The performance remains the same if finetuned on COCO and evaluated on Flickr30K. This concludes that the model trained with our curation method also has stronger generalization ability.

| Flickr30k -> COCO |       B4 |     CIDEr |
|------------------------|---------:|----------:|
| BLIP                   |     31.8 |     108.2 |
| +ReplaceImg (40%)       | **32.3** | **110.2** |
| BEiT-3                 |     21.0 |      76.4 |
| +ReplaceImg (40%)       |     **24.0** |     **85.0** |


| COCO -> Flickr30K |   B4 | CIDEr |
|------------------------|-----:|------:|
| BLIP                   | 25.6 |  67.9 |
| +ReplaceImg (10%)       | 25.6 |  67.8 |
| BEiT-3                 | 25.5 |  67.0 |
| +ReplaceImg (10%)       | 24.7 |  66.9 |

More detailed experiment results are in the Table 1 and Table 2 in the pdf attached below.

---

### Decision · Program_Chairs · 2023-09-21

**Decision:**

Reject

**Comment:**

This paper proposes to improve vision language models by replacing images to produce better image-text alignment. Reviewers agreed that the idea was interesting, but universally felt that the discovered improvement was not large enough to merit acceptance, notably for the replace image method.